# Heavy Labels Out! Dataset Distillation with Label Space Lightening

## Abstract

Dataset distillation or condensation aims to condense a large-scale training dataset into a much smaller synthetic one such that the training performance of distilled and original sets on neural networks are similar. Although the number of training samples can be reduced substantially, current state-of-the-art methods heavily rely on enormous soft labels to achieve satisfactory performance. As a result, the required storage can be comparable even to original datasets, especially for large-scale ones. To solve this problem, instead of storing these heavy labels, we propose a novel label-lightening framework termed **HeLlO** aiming at effective image-to-label projectors, with which synthetic labels can be directly generated online from synthetic images. Specifically, to construct such projectors, we leverage prior knowledge in open-source foundation models, *e.g.*, CLIP, and introduce a LoRA-like fine-tuning strategy to mitigate the gap between pre-trained and target distributions, so that original models for soft-label generation can be distilled into a group of low-rank matrices. Moreover, an effective image optimization method is proposed to further mitigate the potential error between the original and distilled label generators. Extensive experiments demonstrate that with only about 0.001% of the original storage required for a complete set of soft labels, we achieve comparable performance to current state-of-the-art dataset distillation methods on large-scale datasets. Our code will be available.

## 1 Introduction

Dataset distillation Wang et al. (2018) is proposed to deal with the issues caused by large-scale datasets, *e.g.*, high computational overhead for training and heavy burden for storage and transmission. It aims to condense a large dataset into a much smaller synthetic one, which preserves the original training performance, so that it can serve as an effective and efficient surrogate to train downstream neural networks. For instance, it has been demonstrated that a network trained with merely 1 synthetic image per class (IPC) can perform well on CIFAR-10 Krizhevsky et al. (2009). However, with such a high compression ratio, it is challenging for the distilled sets to encapsulate the whole knowledge of the original dataset used for training in a very limited space. Thus, classic methods in this field like Wang et al. (2018); Zhao et al. (2020); Zhao & Bilen (2021; 2023) still have a significant performance gap between the original set and the synthetic one, especially when handling large-scale datasets Yu et al. (2023).

To compensate for such dramatic information loss, recent state-of-the-art dataset distillation methods Shao et al. (2024); Sun et al. (2024); Yin et al. (2024) turn to data augmentation, to make the best use of the limited synthetic data. Specifically, strategies such as Mixup Zhang et al. (2017) and Cutmix Yun et al. (2019) are applied in downstream network training, which effectively enhance the performance of distilled datasets and scale dataset distillation up to larger and more complex datasets like ImageNet Deng et al. (2009).

Nevertheless, these recent works heavily rely on soft labels generated by a pre-trained teacher model on the original dataset. According to RDED Sun et al. (2024), networks trained with 10 IPC on ImageNet-1k achieve only 15.2% accuracy with categorical hard labels, compared to 42.1% with soft labels. Since each augmented sample corresponds to a distinct soft label, as shown in Fig. 1 (left), there are a number of generated soft labels that far exceeds the basic synthetic samples. Consequently, storage costs for these soft labels are non-negligible, especially for large-scale

Figure 1: The soft label generation part of the current state-of-the-art large-scale dataset distillation (left), and our proposed online lightening image-to-label projector framework (right). For the current state-of-the-art large-scale dataset distillation, for each downstream training epoch, soft labels are generated for each augmented image and stored all the soft labels. For our proposed method, we adopt the open-source foundation models as the base models, which are fixed during the whole training process, and introduce a LoRA-like knowledge transfer method to narrow the gap between the original label space and the target one. We only need to store the low-rank matrices, which significantly reduces the storage costs.

datasets with numerous categories. For example, on ImageNet-1K with 1 IPC, the required storage for distilled images is $\sim$ 15 MB, whereas the storage for soft labels exceeds 572 MB—more than 38 times greater. Furthermore, with 200 IPC, the storage required for soft labels reaches 110 GB, making it even comparable to the original dataset size.

To address the issue of such heavy labels, we propose a novel label-lightening framework termed **He**avy **L**abels **O**ut, or **HeLlO** in short. Fig. 1 (right) illustrates the overall framework of the proposed HeLlO. By creating an effective and lightweight projector from the image to the label space, it reduces the required storage significantly. Specifically, we build the projector upon recent foundation models like CLIP Radford et al. (2021) that has been pre-trained on massive data and can readily adapt to various target datasets. To achieve this, we propose an effective LoRA-like Hu et al. (2021) knowledge transfer method that efficiently transforms the original feature space of CLIP into that of the target data. As an efficient alternative to the teacher model trained on the target dataset for soft-label generation, the derived low-rank matrices can be seen as a transferable and lightweight representation for the original label space.

Interestingly, by leveraging the vision-language alignment capability in CLIP Zhang et al. (2022), we propose initializing the projector with the textual representation of label categories, providing a strong starting point that improves training and convergence. Moreover, we propose an effective image optimization method to further reduce the potential error between the original and distilled label generators. Our extensive experiments show that with only 0.001% of the original storage for soft labels, we achieve performance comparable to, or even better than, state-of-the-art large-scale dataset distillation methods.

In summary, our contributions are as follows:

- We are the first to focus on the issue of heavy labels in dataset distillation to our best knowledge and propose an effective label-lightening framework termed HeLlO to address the problem.
- By leveraging pre-trained CLIP, the proposed HeLlO method compresses the storage of massive soft labels into a set of lightweight low-rank matrices and tailors an initialization method based on CLIP's textual representation to enhance optimization.
- We introduce an image-level optimization technique that further minimizes the gap between the original and distilled label generators.
- Extensive experiments validate the comparable or even superior performance to state of the arts using just 0.001% of the storage required for synthetic labels.

## 2 RELATED WORKS

Dataset distillation or condensation Wang et al. (2018); Zhao et al. (2020); Yu et al. (2023) aims to solve the issues of massive storage, transmission burden, and computational costs for downstream tasks caused by large-scale datasets. Specifically, it condenses the whole knowledge of the original large-scale datasets into a much smaller space and preserves the performance. The mainstream dataset distillation methods can be roughly classified into three categories, according to their optimization objectives: performance matching Wang et al. (2018); Deng & Russakovsky (2022); Loo et al. (2022); Zhou et al. (2022); Nguyen et al. (2020; 2021), parameter matching Zhao et al. (2020); Zhao & Bilen (2021); Cazenavette et al. (2022); Cui et al. (2023); Du et al. (2023); Guo et al. (2023); Liu et al. (2022a) and distribution matching Zhao & Bilen (2023); Wang et al. (2022); Zhao et al. (2023); Sajedi et al. (2023).

Traditional dataset distillation methods suffer scaling-up problems due to the bi-level optimization problems, such that the gradients should backpropagate through an unrolled computation graph Yu et al. (2023). Recent work SRe$^2$L Yin et al. (2024) proposes a variant distribution matching paradigm to decouple the bi-level optimization and scales up to the full-size ImageNet-1K dataset. It matches the distribution in feature space of the synthetic dataset and the statistical information of the original dataset stored in the batch normalization layers of the pre-trained model. Further, G_VBSM Shao et al. (2024) utilize multiple pre-trained teachers to provide more statistical information and improve the transferability across different architectures. RDED Sun et al. (2024) is the current state-of-the-art large-scale dataset distillation method, which is based on selection instead of synthesizing. It selects and concatenates the most representative patches evaluated by the pre-trained teacher model.

However, due to the significant reduction in the size of the datasets, an apparent performance gap still exists between the original dataset and the distilled one. For small-scale dataset distillation, a series of works Bohdal et al. (2020); Sucholutsky & Schonlau (2021); Cui et al. (2023); Deng & Russakovsky (2022); Nguyen et al. (2021); Loo et al. (2022); Zhou et al. (2022) expand the label space by transforming the one-hot labels to soft labels, which apparently improve the performance for downstream tasks and also provides a new perspective to condense dataset comprehensively. However, simply transforming the one-hot label to a soft label for each synthetic image is not effective for large-scale dataset distillation, as the plain soft labels do not provide sufficient extra knowledge for downstream tasks. In order to solve this issue and compensate for the huge reduction in the number of data, current large-scale dataset distillation methods Shao et al. (2024); Sun et al. (2024); Yin et al. (2024) adopt the extensive data augmentation strategies, *e.g.*, Mixup Zhang et al. (2017) and Cutmix Yun et al. (2019), and generate soft labels for each augmented image. It will increase the diversity of the distilled data for downstream training, and significantly improve the performance for downstream tasks. However, generating such labels requires restoring huge amount of soft labels, and for large-scale datasets, it will cause non-negligible storage costs. Focusing on this issue, our proposed method only requires 0.001% storage space while obtaining comparable performance with the state-of-the-art large-scale dataset distillation methods.

## 3 METHODS

### 3.1 PRELIMINARY

For the large-scale dataset $\mathcal{T} = (X_t, Y_t)$, where $X_t \in \mathbb{R}^{N_t \times D}$ and $Y_t \in \mathbb{R}^{N_t \times C}$, dataset distillation aims to learn a much smaller dataset $\mathcal{S} = (X_s, Y_s)$, where where $X_s \in \mathbb{R}^{N_s \times D}$ and $Y_s \in \mathbb{R}^{N_s \times C}$, such that the models train on both two datasets can obtain similar performance. Here, $N_t$ and $N_s$ refer to the number of samples in $\mathcal{T}$ and $\mathcal{S}$, $N_t \gg N_s$, and $D$ and $C$ are the dimension of the images and labels. Current state-of-the-art large-scale dataset distillation methods Shao et al. (2024); Sun et al. (2024); Yin et al. (2024) all follow the teacher(s)-guided soft label generation strategy. It generates a soft label for each augmented image, and the label space is expanded to $\mathbb{R}^{K \times N_s \times C}$, where $K$ is the number of training iterations for downstream tasks. It can be formulated as follows:

$$X_s^* = \arg\min_{X_s} \mathcal{L}(\mathcal{S}, \mathcal{T}),$$
$$Y_s^* = \frac{1}{|\Theta_{\mathcal{T}}|} \sum f_{\theta \sim \Theta_{\mathcal{T}}}(\mathcal{A}(X_s^*)), \tag{1}$$

where $\mathcal{L}$ is the optimization objectives to update the distilled images, $\Theta_{\mathcal{T}}$ refers to teacher model(s) ($|\Theta_{\mathcal{T}}| \geq 1$), and $\mathcal{A}$ is the augmentation methods. Each augmented distilled image requires generating the corresponding soft labels, which will cause a huge storage burden.

## 3.2 EFFICIENT INITIALIZATION OF SURROGATE PROJECTION

To effectively and efficiently transfer the label space in a lightweight way and easily adapt it to different datasets, we adopt the open-source and pre-trained foundation model, CLIP, as our base model. It does not require extra storage space and can be accessed on demand. Specifically, we adopt the paradigm of linear probe CLIP by utilizing the image encoder part of CLIP and following with a linear transformation. The image encoder of CLIP is pre-trained on numerous paired data and can provide accurate and knowledge-rich features, which makes the linear probe CLIP a powerful classifier. Here, the parameters required to store is only the linear transformation part.

However, the storage cost of the linear transformation part depends on the number of classes of the original dataset, which will be non-negligible for large-scale datasets with a large number of classes. Also, there still exists a gap between the original label space and the lightening one, which may make transferring to downstream tasks difficult. Here, to reduce the storage cost for the linear transformation part and improve the ability to transfer, we propose a novel storage-efficient initialization strategy. Here, given a pre-trained multi-modal foundation model, *e.g.*, CLIP, we denote the image encoder part as $\mathcal{E}_I$ and the text part as $\mathcal{E}_T$. For any dataset $\mathcal{D} = (X, Y)$, we can simply obtain the text descriptions $R = \{r^{(i)}\}_{i=0}^{C-1}$ for the whole dataset by utilizing the vanilla prompt engineering technique Radford et al. (2021) with fixed templates. We adopt the fixed normalized text embedding of the descriptions for all classes as the initialization of the linear transformation, which significantly saves storage space as we do not need to store the initial parameters. Also, it can improve the basic performance of our label projector as the proposed initialization is equivalent to the pre-trained zero-shot classification. Following we will provide the theoretical analysis.

**Proposition 1.** *Text embedding initialized linear transformation is equivalent to the pre-trained zero-shot classification.*

*Proof.* For basic zero-shot CLIP prediction, we have:

$$c^* = \underset{i \in \{0, \ldots, C-1\}}{\arg\max} \ \mathcal{S}im(x, r^{(i)}),$$

$$\mathcal{S}im(x, r^{(i)}) = v_I \cdot (v_T^{(i)})^T, where \quad (2)$$

$$v_I = \frac{\mathcal{E}_I(x)}{||\mathcal{E}_I(x)||}, v_T^{(i)} = \frac{\mathcal{E}_T(r^{(i)})}{||\mathcal{E}_T(r^{(i)})||},$$

where $x$ refers to the input image(s), $r^{(i)}$ is the text description for class $i$, and $v_I \in \mathbb{R}^{B \times d_f}$ and $v_T \in \mathbb{R}^{C \times d_f}$ refer to the normalized embedding for the input image and the text description, $v_T^{(i)}$ is for $i^{th}$ class. $B$ is the batch size of the input image(s), and $d_f$ is the dimension of the output embedding. As for linear probe one, denote the parameters of the linear transformation is $W \in \mathbb{R}^{d_f \times C}$, $W = [w^{(0)}, w^{(1)}, \ldots, w^{(C-1)}]$, and here, numerically, $W = (v_T)^T$ and $w^{(i)} = (v_T^{(i)})^T$. The classification can be formally written as:

$$c^* = \underset{i \in \{0, \ldots, C-1\}}{\arg\max} \ v_I \cdot w^{(i)} + b, where \ w^{(i)} = (v_T^{(i)})^T. \quad (3)$$

Here, we set the bias $b$ zero, and these two operations are equivalent. $\square$

## 3.3 LoRA-LIKE LOW-RANK KNOWLEDGE TRANSFER

As mentioned before, we adopt the fixed initialization for the linear transformation part, which will not introduce any extra storage costs and can improve the basic classification ability of the linear probe CLIP. However, there still exists a significant gap between the original label space and the lightening one, which may cause difficulties transferring to downstream tasks. Here, one typical way to solve the above issues is fine-tuning the whole projector to the target label space, but it requires huge extra computational costs to train the complex foundation model and non-negligible storage space to save the tuned parameters.

In order to reduce the computational costs and the storage costs, while narrowing the gap and further improving the transferability of the projector to the downstream tasks, we propose a novel parameter-efficient knowledge transfer method. First of all, to minimize the cost of fine-tuning, we follow the idea of LoRA Hu et al. (2021), which decomposes the weight matrix of the foundation models into low-rank matrices. It will preserve the pre-trained knowledge and enhance efficiency by reducing the number of updated parameters. Formally, for specific fine-tuning target $\mathcal{L}$, we have:

$$\theta^* = \arg\min_{\theta} \mathcal{L}(\mathcal{D}; \theta), where$$
$$\theta^* = \theta_0 + \Delta\theta, \ \Delta\theta = A \cdot B. \tag{4}$$

Here, $\mathcal{D}$ refers to the target dataset, $\theta_0 \in \mathbb{R}^{d \times k}$ is the initial pre-trained parameters of the model, and $\Delta\theta$ is the incremented weight, which is updated during the fine-tuning procedure. $A \in \mathbb{R}^{d \times r}$ and $B \in \mathbb{R}^{r \times k}$ are the decomposed low-rank matrices of $\Delta\theta$, where $r \ll d$ and $r \ll k$, largely relieving the computational and storage burden. Specifically, we apply LoRA to both the image encoder and the linear transformation parts (while with different ranks), avoiding fine-tuning the whole model and saving storage space. Moreover, to further improve the transferability to the downstream tasks, we combine the original LoRA target optimization objective with the multi-teacher knowledge transfer metric as follows:

$$\mathcal{L}(\mathcal{D}; \theta) = MSE(f_\theta(X), Y') + \lambda CE(f_\theta(x), Y). \tag{5}$$

Here, $\theta$ is the parameters of the projector, $f_\theta(X) = \mathcal{E}_I(X)W$, and $Y'$ refers to the soft labels generated by the weak teachers $\Theta'_{\mathcal{T}}$, such that $Y' = \frac{1}{|\Theta'_{\mathcal{T}}|} \sum f_{\theta \sim \Theta'_{\mathcal{T}}}(X)$. Here, weak teacher is the model trained on the original dataset terminated at the early training stage. Practically, we adopt the original dataset $\mathcal{T}$ as the target dataset, and weak teachers are from the early stage of the single training trajectory for easy to obtain and transfer.

## 3.4 Synthetic Dataset Initialization and Update

Here, we follow RDED Sun et al. (2024), to initialize the distilled dataset $\mathcal{S}$. Specifically, as the image patches can effectively represent object features, they select patches based on their difficulty and concatenate the patches to form an image. Specifically, they adopt the teacher model $\theta_t$ as the observer to evaluate the difficulty of the patches, and the most representative patches will be selected. The selection metric is as follows:

$$p^* = \arg\min_{p \sim \mathcal{P}} CE(f_{\theta_t}(p), y_p), \tag{6}$$

where $\mathcal{P}$ is a bunch of patches random cropped from the images of the original dataset $\mathcal{T}$, and $y_p$ is the corresponding labels of the original image. However, to reduce storage costs, we propose a surrogate parameter-efficient model to replace the original teacher model. This substitution introduced a performance gap, as the observer model is not the projector model for the downstream tasks. To narrow this gap, we further update the synthetic dataset to minimize the information loss of patches on the surrogate projector. Here, we follow LIC Anonymous (2024) to do the image update, and the adapted optimization metric is as follows:

$$\mathcal{G}(\mathcal{E}_I, p) = MSE(\mathcal{E}_I(p), \mathcal{E}_I(\hat{p})), \tag{7}$$

where $\hat{p}$ is the transformed one with first down-sampled and then up-sampled to the original size. It will further reduce the information loss on the projector, and narrow the performance gap between the observer and the projector.

## 3.5 Algorithm Summary

In summary, we propose a novel label-lightening framework, HeLlO, building an effective and efficient image-to-label projection with lower storage requirements. The framework of HeLlO is shown in Algorithm 1. Here, we first initialize the synthetic dataset $\mathcal{S}$ with the metric Eq. 6, which selects the most representative patches of the dataset. Then, we initialize the linear transform part using the normalized text embedding, generated by the fixed text descriptions and the pre-trained text encoder without any extra storage space. Following we adopt the LoRA-like knowledge transfer method Eq. 5 to efficiently fine-tune the projector with weak teachers' guidance, and this step

---

**Algorithm 1** HeLlO Framework

---

1: **Input:** Original dataset $\mathcal{T}$, open-source model $\theta$, weak teachers $\Theta_{\mathcal{T}}$;
2: **Output:** Synthetic dataset $\mathcal{S}$;
3: Initialize $\mathcal{S}$ with difficulty evaluation Eq. 6;
4: Generate normalized text embedding with text descriptions $R = \{r^{(i)}\}_{i=0}^{C-1}$, $v_T^{(i)} = \frac{\mathcal{E}_T(r^{(i)})}{||\mathcal{E}_T(r^{(i)})||}$;
5: Initialize the linear transformation part with normalized text embedding, $W = (v_T)^T$;
6: **repeat**
7:     Update incremented parameters $\Delta\theta = A \cdot B$ with low-rank knowledge transfer Eq. 5;
8: **until** Convergence
9: **repeat**                                                ▷ Optional
10:     Update images using Eq. 7;
11: **until** Convergence
12: **for** $e < K$ **do**     ▷ For online image-to-label projecting during downstream task training
13:     $Y^* = f_\theta(\mathcal{A}(X_s))$;
14:     $\phi^e = \phi^{e-1} - \alpha\nabla_\phi(MSE(f_\phi(\mathcal{A}(X_s)), Y^*) + \beta CE(f_\phi(\mathcal{A}(X_s)), Y_s))$     ▷ $\mathcal{A}$ is the augmentation method, and $\phi$ refers to the parameters of the student model
15: **end for**

---

will only cause very low storage costs. As we use the projector to replace the observer model to relabel the images for downstream training, there exists a performance gap. To further narrow this gap and reduce the information loss on the projector model, we adopt Eq. 7 to update the synthetic data. Lastly, for the downstream training, the synthetic labels can be directly generated online from synthetic images through the projector.

## 4 EXPERIMENTS

### 4.1 EXPERIMENT SETTING

#### 4.1.1 DATASETS AND NETWORKS

Our proposed method HeLlO aims to solve the heavy-label issue in the large-scale dataset distillation methods. Here, we adopt the high-resolution datasets ImageNet-100, Places365-Standard Zhou et al. (2017) and ImageNet-1K Deng et al. (2009) as the validation datasets to show the efficacy of our proposed method. All of these datasets are $224 \times 224$ in size.

As for networks, we adopt CLIP (ResNet-50) from the official Open-AI as the base model, followed by a linear transformation. For baseline comparison, we follow the prior works Yin et al. (2024); Shao et al. (2024); Sun et al. (2024), adopting ResNet-18 He et al. (2016) as the evaluation model. Also, to show the generalization ability across various architectures of our proposed method, we select ShuffleNet-V2 Ma et al. (2018), MobileNet-V2 Sandler et al. (2018), EfficientNet-B0 Tan & Le (2019), Swin-V2-Tiny Liu et al. (2022b), and VGG-11 Simonyan (2014) as the evaluation architectures.

#### 4.1.2 IMPLEMENTATION DETAILS

For surrogate projector training, we first initialize the linear transformation part with text embedding. We use the official prompt engineering templates provided by the CLIP code base to generate the text description and use the text encoder (from official CLIP with ResNet-50) to generate the text embedding. During the training process, we propose a LoRA-like knowledge transfer method to further improve the transferability of our method to the downstream tasks. Here, we efficiently fine-tune the convolution layer in the image encoder part and the linear transformation part. Specifically, for ImageNet-100, we use rank 8 for the image encoder part, and 64 for the linear transformation part. For both Places365-Standard and ImageNet-1K, we use rank 8 for the image encoder part, and 128 for the linear transformation part. We also utilize multi-weak teachers as guidance to generate the soft labels for projector learning. In practice, we train a ResNet-18 model from scratch using the PyTorch official code base and select some checkpoints along the training trajectory. The teachers

Table 1: Comparison with baseline methods. $^*$ indicates the evaluation results reproduced by us, **bold** refers to the best results and underline refers to the second best results. Here, all methods adopt ResNet-18 as the evaluation model. Here, the Actual Extra Storage refers to the extra storage required for downstream tasks for IPC 1, 10, and 50 (for teacher models (RDED) or the soft labels and augmentation information (for SRe$^2$L and G_VBSM, for 300 epochs)).

| Datasets | | SRe$^2$L Yin et al. (2024) | G_VBSM Shao et al. (2024) | RDED Sun et al. (2024) | Ours |
|---|---|---|---|---|---|
| **ImageNet-100** | 1 | $3.0 \pm 0.3$ | - | $8.1 \pm 0.3$ | **$12.5 \pm 0.2$ (+ 4.4)** |
| | 10 | $9.5 \pm 0.4$ | - | $36.0 \pm 0.3$ | **$48.9 \pm 0.1$ (+ 12.9)** |
| | 50 | $27.0 \pm 0.4$ | - | $61.6 \pm 0.1$ | **$69.4 \pm 0.1$ (+ 7.8)** |
| | **Actual Extra Storage** | [6.9MB, 64.8MB, 324.2MB] | - | [42.8MB, 42.8MB, 42.8MB] | **[2.6MB, 2.6MB, 2.6MB]** |
| **Places365-Standard** | 1 | $1.4 \pm 0.2^*$ | - | $5.0 \pm 0.1^*$ | **$7.1 \pm 0.1$ (+ 2.1)** |
| | 10 | $9.5 \pm 0.1^*$ | - | $29.2 \pm 0.1^*$ | **$33.4 \pm 0.3$ (+ 4.2)** |
| | 50 | $31.3 \pm 0.1^*$ | - | **$44.0 \pm 0.1^*$** | $41.2 \pm 0.1$ (-) |
| | **Actual Extra Storage** | [79.3MB, 790.4MB, 3950.6MB] | - | [43.4MB, 43.4MB, 43.4MB] | **[3.0MB, 3.0MB, 3.0MB]** |
| **ImageNet-1K** | 1 | $0.1 \pm 0.1$ | $1.7 \pm 0.1^*$ | $6.6 \pm 0.2$ | **$12.9 \pm 0.3$ (+ 6.3)** |
| | 10 | $21.3 \pm 0.6$ | $31.4 \pm 0.5$ | $42.0 \pm 0.1$ | **$43.7 \pm 0.1$ (+ 1.7)** |
| | 50 | $46.8 \pm 0.2$ | $51.8 \pm 0.4$ | **$56.5 \pm 0.1$** | $52.2 \pm 0.1$ (-) |
| | **Actual Extra Storage** | [579.8MB, 5798.3MB, 28990.8MB] | [582.2MB, 5821.5MB, 29110.6MB] | [44.7MB, 44.7MB, 44.7MB] | **[3.3MB, 3.3MB, 3.3MB]** |

Table 2: Evaluation results of cross-architecture generalization under the ImageNet-100, Places365-Standard and ImageNet-1K with IPC 10. $^*$ indicates the evaluation results reproduced by us.

| Datasets | | ShuffleNet-V2 | MobileNet-V2 | EfficientNet-B0 | Swin-V2-Tiny | VGG-11 |
|---|---|---|---|---|---|---|
| **ImageNet-100** | **RDED** | $27.7 \pm 0.6^*$ | $35.7 \pm 0.3^*$ | $37.9 \pm 0.1^*$ | $18.0 \pm 0.1^*$ | $21.2 \pm 0.4^*$ |
| | **Ours** | **$32.7 \pm 0.8$ (+ 5.0)** | **$40.6 \pm 0.8$ (+ 4.9)** | **$47.0 \pm 0.1$ (+ 9.1)** | **$24.2 \pm 0.3$ (+ 6.2)** | **$27.6 \pm 0.1$ (+ 6.4)** |
| **Places365-Standard** | **RDED** | $18.1 \pm 0.7^*$ | $21.0 \pm 0.5^*$ | $26.3 \pm 0.1^*$ | $14.0 \pm 0.2^*$ | $12.8 \pm 0.1^*$ |
| | **Ours** | **$22.1 \pm 0.6$ (+ 4.0)** | **$27.3 \pm 0.2$ (+ 6.3)** | **$31.9 \pm 0.3$ (+ 5.6)** | **$18.1 \pm 0.3$ (+ 4.1)** | **$17.7 \pm 0.3$ (+ 4.9)** |
| **ImageNet-1K** | **RDED** | $23.3 \pm 0.1^*$ | $34.4 \pm 0.2$ | $42.8 \pm 0.5$ | $17.8 \pm 0.1$ | $22.7 \pm 0.1$ |
| | **Ours** | **$26.5 \pm 0.2$ (+ 3.2)** | **$38.1 \pm 0.5$ (+ 3.7)** | **$44.4 \pm 0.2$ (+ 1.6)** | **$29.5 \pm 0.1$ (+ 11.7)** | **$24.2 \pm 0.3$ (+ 1.5)** |

are in different stages for different IPCs, and we use 9 teachers for projector training. For more implementation details, please refer to the supplementary materials.

## 4.2 RESULTS ON BASELINES

Our method aims to solve the heavy-label issues in the large-scale dataset distillation methods. Here, we compare our proposed method with prior state-of-the-art large-scale dataset distillation methods, SRe$^2$L Yin et al. (2024), G_VBSM Shao et al. (2024), and RDED Sun et al. (2024). Following the experiment setting with previous works and fair comparison, we use the distilled dataset to train several random initialized ResNet-18 from scratch, and the evaluation results are reported in Table 1. From the results, our proposed method only requires very low storage space costs for label generation that can get comparable performance. Particularly for the smaller distilled dataset generation (smaller IPCs or classes), our proposed method demonstrates superior performance, achieving state-of-the-art results that exceed those of previous methods by a remarkable margin of up to 12.9% under the setting of ImageNet-100 with IPC 10. Moreover, it accomplishes this while simultaneously siginificantly reducing associated storage costs.

## 4.3 RESULTS ON CROSS-ARCHITECTURE GENERALIZATION

The ability to generalize to different architectures is an important standard to measure the performance of the distilled dataset, which shows the practicality to the downstream tasks. Here, we evaluate the cross-architecture performance of the previous state-of-the-art method RDED and our proposed method on the ImageNet-100, Places365-Standard and the ImageNet-1K with IPC 10. We adopt five different architectures ShuffleNet-V2, MobileNet-V2, EfficientNet-B0, Swin-V2-Tiny, and VGG-11. The results are shown in Table 2. From the results, our proposed method demonstrates state-of-the-art performance across various architectures. For residual-like architectures (ShuffleNet-V2, MobileNet-V2, and EfficientNet-B0), and convolutional networks (VGG-11), our proposed method shows a superior transferability than the previous state-of-the-art method RDED. Surprisingly, our method demonstrates exceptional transferability on transformer architectures, surpassing the previous state-of-the-art by an impressive margin of 6.2% (ImageNet-100),

Table 3: The results of the ablation studies for the effectiveness of each step of our proposed method. From left to right, each step is incremented based on the former one. The experiments are under the settings of ImageNet-1K with IPC 1 and 10.

| | Probe Linear CLIP | + Multi-Weak-Teacher Guided | + LoRA-Like Knowledge Transfer | + Text-Embedding-Based Init. | + Image Update |
|---|---|---|---|---|---|
| **Acc.-IPC 1** | 5.3 ± 0.1 | 6.4 ± 0.2 (+ 1.1) | 11.4 ± 0.2 (+ 5.0) | 11.9 ± 0.1 (+ 0.5) | 12.9 ± 0.3 (+ 1.0) |
| **Acc.-IPC 10** | 28.2 ± 0.2 | 30.1 ± 0.1 (+ 1.9) | 43.5 ± 0.1 (+ 13.4) | 43.6 ± 0.1 (+ 0.1) | 43.7 ± 0.1 (+ 0.1) |
| **#Params** | 1.0M | 1.0M (-) | 1.5M (↑ 0.5) | 0.8M (↓ 0.7) | 0.8M (-) |

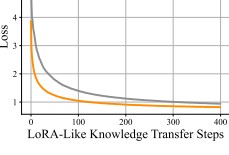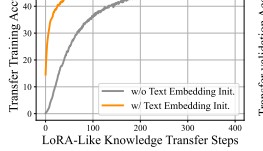

Figure 2: The trends of the loss, transfer training accuracy, and transfer validation accuracy during the LoRA-Like Knowledge Transfer process. Here, the experiments is under the setting of IPC 10 for ImageNet-1K.

Table 4: The results of the ablation studies for the impact of the different learnable parameters in LoRA-like transfer learning (left), and the different stages of teachers (right). The experiments are conducted under the setting of IPC 10 for ImageNet-100 and ImageNet-1K.

| | 0.3M / 0.4M | 0.6M / 0.8M | 0.9M / 1.2M | 1.2M / 1.6M | 1-41 | 11-51 | 21-61 | 31-71 | 41-81 | 51-91 |
|---|---|---|---|---|---|---|---|---|---|---|
| **ImageNet-100** | 45.4 ± 0.2 | 48.9 ± 0.1 | 49.7 ± 0.1 | 49.8 ± 0.2 | 48.9 ± 0.1 | 47.5 ± 0.1 | 45.5 ± 0.1 | 44.2 ± 0.1 | 43.9 ± 0.7 | 41.5 ± 0.4 |
| **ImageNet-1K** | 38.8 ± 0.1 | 43.7 ± 0.1 | 44.2 ± 0.1 | 44.4 ± 0.1 | 42.3 ± 0.1 | 43.7 ± 0.1 | 42.1 ± 0.1 | 41.2 ± 0.1 | 40.6 ± 0.1 | 39.7 ± 0.3 |

4.1% (Places365-Standard) and 11.7% (ImageNet-1K) on Swin-V2-Tiny, while only requires very low storage costs for the label generation.

## 4.4 ABLATION STUDY

### 4.4.1 THE IMPACT OF KEY FACTORS

To validate the effectiveness of our proposed method, we designed a series of ablation experiments to evaluate each component of our method. The results are shown in Table 3. Here, we start from the plain linear probe CLIP. We directly use the original dataset to train the linear probe CLIP and use it to online generate the labels during the downstream tasks training. As the results shown in Table 3, it only obtains 5.3% and 28.2% accuracy for IPC 1 and IPC 10, while requiring 1.0M parameters to store. Based on that, we adopt multi-weak teachers to guide the linear probe CLIP training, which gains 1.1% and 1.9% improvement and maintains the storage costs. Then, we introduce the LoRA-like knowledge transfer method, which significantly improves the performance of downstream training by 5.0% and 13.4% but causes an increase in storage. Following we propose the text-embedding-based initialization strategy, such that we do not need to store the whole linear transformation part but the low-rank matrices. It helps largely reduce the storage costs by 0.8M while maintaining the performance. It also accelerates the convergence speed of the process of the LoRA-Like Knowledge Transfer, as shown in Fig 2. Lastly, we narrow the gap of the original distribution and the target one by updating the images, which improves the performance of the distilled dataset by 1.0% on ImageNet-1K with IPC 1.

### 4.4.2 THE IMPACT OF DIFFERENT RANK

We also explore the impact of ranks of the low-rank matrices in the LoRA-like knowledge transfer part. It also reflects the relation between the number of learnable parameters and the performance. The results are shown in Table 4 (left). From the results, we find that the ranks of the low-rank matrices or the number of learnable parameters can significantly influence the performance of the

Table 5: The downstream accuracy for different architectures of the projector. Here, "CLIP-LoRA (Ours)" and "DINOv2-LoRA (Ours)" refer to the foundation models CLIP and DINOv2 with our proposed methods. ResNet-18, ResNet-50, ViT_b_16 are pre-trained on the ImageNet-1K, from the official torchvision codebase. "CLIP" and "DINOv2" mean directly apply the pre-trained foundation models CLIP and DINOv2 as the label projectors for downstream training. The experiments are under the setting of ImageNet-1K with IPC 10.

| | CLIP-LoRA (Ours) | DINOv2-LoRA (Ours) | ResNet-18 | ResNet-50 | ViT-B-16 | CLIP | DINOv2 |
|---|---|---|---|---|---|---|---|
| Downstream Accuracy | $43.7 \pm 0.1$ | $44.2 \pm 0.1$ | $42.0 \pm 0.1$ | $34.9 \pm 0.1$ | $17.8 \pm 0.2$ | $28.2 \pm 0.2$ | $23.3 \pm 0.2$ |

Table 6: Comparison with logits quantization strategies. The experiments are under the setting of ImageNet-1K with IPC 10.

| | FP32 | FP16 | INT8 | INT4 | Ours |
|---|---|---|---|---|---|
| Downstream Accuracy | $42.0 \pm 0.1$ | $42.2 \pm 0.2$ | $41.2 \pm 0.1$ | $39.3 \pm 0.1$ | $43.7 \pm 0.1$ |
| Memory Costs | 11596.6MB | 5798.3MB (0.5×) | 2899.2MB (0.25×) | 1449.6MB (0.125×) | 3.3MB (3e-4×) |

downstream tasks. However, this effect is pronounced only when the number of learnable parameters is insufficient; once a sufficient level is reached, further increases in learnable parameters do not lead to notable improvements in performance. The inflection point in the results occurs at 0.6M/0.8M for ImageNet-100 and ImageNet-1K. This also indicates that our method is robust to the selection of the ranks; as long as ranks reach a sufficient level, the results remain stable without significant fluctuations.

### 4.4.3 THE IMPACT OF DIFFERENT STAGES OF TEACHERS

In our proposed method, we adopt multi-weak teachers to guide the projector training. Here, we explore the impact of the stage of the teachers on the performance of the downstream tasks. Here, the experiments are under the setting of IPC 10 for both ImageNet-100 and ImageNet-1K. The results are shown in Table 4 (right). The results indicate that the stage of the teachers has a particularly significant impact on the performance of the downstream tasks. For smaller IPCs, earlier-stage teachers are more beneficial for transferring to downstream tasks. In contrast, later-stage teachers tend to contain more complex knowledge that is difficult to decouple and learn effectively.

### 4.4.4 THE IMPACT OF THE ARCHITECTURES OF THE PROJECTOR

In our proposed method, we adopt the foundation model CLIP to validate the effectiveness of our proposed method. We would like to note that our proposed method is also compatible with other foundation models, such as DINOv2 Oquab et al. (2023), which can serve as an alternative to CLIP for LoRA-like knowledge transfer and related strategies. As indicated in Table 5, our proposed method remains highly effective when applied to other foundation models. Meanwhile, we also experiment with directly applying pre-trained models of other architectures as projectors, *e.g.*, ResNet-50, ViT-B-16. These models require significantly more storage space and show much poorer performance on downstream tasks.

### 4.5 COMPARISON WITH LOGITS QUANTIZATION STRATEGIES

To demonstrate the effectiveness and efficiency of our proposed method, we conduct experiments under the ImageNet-1K IPC 10 setting on an NVIDIA RTX A5000 GPU. In these experiments, we apply precision reduction to the labels for the previous state-of-the-art method and compare the results to those obtained with our proposed method. From the results in Table 6, it can be observed that reducing the precision to FP16 (50%) or INT8 (25%) does not lead to a significant decline in downstream performance. However, a noticeable decline occurs when the precision is further reduced to INT4 (12.5%), which indicates the limit for logit quantization. Despite this, our

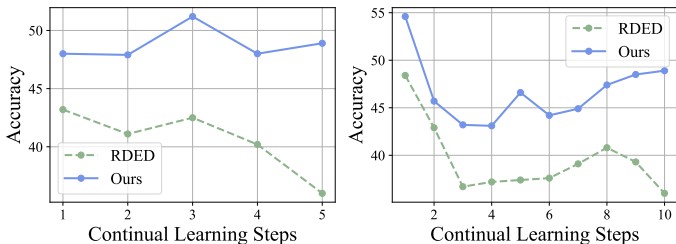

Figure 3: The results on the continual learning for 5-step (left) and 10-step (right). All experiments are conducted under the setting of IPC 10 for ImageNet-100.

Table 7: Results on the aerial dataset AID with IPC 1, 5, and 10. Here, the required extra storage costs for RDED is the actual storage space for the teacher model.

| Downstream Accuracy | IPC 1 | IPC 5 | IPC 10 | Required Extra Storage Costs |
|---|---|---|---|---|
| **RDED** | $27.2 \pm 0.8$ | $61.1 \pm 0.3$ | $78.9 \pm 0.2$ | 42.69MB |
| **Ours** | $\mathbf{28.8 \pm 0.5}$ | $\mathbf{65.6 \pm 0.8}$ | $\mathbf{79.2 \pm 0.2}$ | **2.8MB (0.065× of the original extra storage)** |

proposed method achieves comparable performance while compressing storage requirements to less than 0.1%.

### 4.6 RESULTS ON PRACTICAL DATASETS

To demonstrate the generalizability and effectiveness of our proposed method, we also conduct experiments on the aerial dataset AID Xia et al. (2017). These experiments are performed under the AID settings with IPC values of 1, 5, and 10. The performance results, along with the associated extra storage costs, are presented in the Table 7 (where we adopt an online soft label generation strategy and consider the teacher model as the extra storage cost). The results indicate that our proposed method is effective across different high-resolution datasets.

### 4.7 RESULTS ON CONTINUAL LEARNING

Continual learning De Lange et al. (2021); Wang et al. (2024); Rebuffi et al. (2017) is an important application for dataset distillation Yu et al. (2023). Here, for fair comparison, we follow the previous works Zhao & Bilen (2023); Yin et al. (2024), adopting the GDumb Prabhu et al. (2020) framework to evaluate the performance on continual learning. The experiments are conducted under the setting of ImageNet-100 with IPC 10, and we evaluate both the 5-step and the 10-step settings. The results are shown in Fig. 3. From the results, our proposed method is significantly superior to the previous state-of-the-art method RDED.

## 5 CONCLUSION

In this paper, we propose a novel label-lightening framework termed HeLlO, aiming to solve the heavy-label issue in large-scale dataset distillation. Our method involves an effective image-to-label projector, with which the synthetic labels can be directly generated online from synthetic images during training downstream networks. Specifically, we leverage the prior knowledge in open-source foundation models and introduce a parameter-efficient LoRA-like fine-tuning method to narrow the gap between the label distribution of the pre-trained and target ones, which improves the transferability of the projector to the downstream tasks as well. Moreover, we propose a text-guided initialization strategy for the projector that enhances training. To further address the gap between the original label generator and the projector, we also develop a strategy to optimize synthetic images within the projector. Extensive experiments demonstrate that the proposed HeLlO achieves performance comparable or even superior to current state-of-the-art dataset distillation techniques while using just about 0.001% of the original label storage space.

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
