# OpenReview forum: "Heavy Labels Out! Dataset Distillation with Label Space Lightening"
_ICLR.cc/2025/Conference — Submitted to ICLR 2025_

### Official Review · Reviewer_1rVR · 2024-10-29

**Soundness:** 2
**Presentation:** 3
**Contribution:** 2
**Rating:** 6
**Confidence:** 4

**Summary:**

Dataset distillation aims to condense large training datasets into smaller synthetic ones, but current methods still demand substantial storage for soft labels. The proposed method, HeLlO, tackles this issue by generating synthetic labels directly from images using foundation models like CLIP and fine-tuning with low-rank matrices, significantly cutting down storage requirements. This approach delivers performance on par with state-of-the-art methods while using just 0.003% of the original storage.

**Strengths:**

[S1] This paper tackles one of the key challenges in efficient training and dataset distillation (DD).

[S2] The writing is clear and easy to follow.

[S3] The results show comparable performance to some of the state-of-the-art methods on large-scale datasets, but the experiments could be more extensive, especially in terms of comparisons and generalizability.

**Weaknesses:**

[W1] There are concerns about the method's generalizability, especially regarding its applicability to high-resolution datasets and various model architectures. I suggest testing DRUPI on 256x256 ImageNet-1k, as well as practical datasets like clinical and aerial images, to better assess its versatility. Additionally, evaluating its performance on ViT models would help determine how well the method generalizes across different architectures.

[W2] An analysis of distillation costs is missing from the paper, which is crucial for assessing the efficiency of data-efficient training algorithms. A detailed comparison of the training costs for both the proposed method and state-of-the-art (SOTA) approaches, especially in terms of memory usage and training time, is necessary.

[W3] The paper lacks comparisons with key state-of-the-art methods such as DREAM, DataDAM, and SeqMatch in Table 2. Adding these comparisons would offer a more complete evaluation of DRUPI's effectiveness. If these methods were excluded for specific reasons, it would be helpful for the authors to explain their omission and clarify how DRUPI compares to these recent approaches.

**References**:
- [a] Liu, Yanqing, et al. "DREAM: Efficient dataset distillation by representative matching." Proceedings of the IEEE/CVF International Conference on Computer Vision. 2023.
- [b] Sajedi, Ahmad, et al. "DataDAM: Efficient dataset distillation with attention matching." Proceedings of the IEEE/CVF International Conference on Computer Vision. 2023.
- [c] Yin, Zeyuan, et al. "Squeeze, recover and relabel: Dataset condensation at ImageNet scale from a new perspective." Advances in Neural Information Processing Systems 36 (2024).

**Questions:**

1. Does the method demonstrate cross-architecture generalization capability on ViT models?
2. Can HeLlO maintain strong performance at higher IPCs, such as 100 or 200?
3. How does the proposed framework perform on other downstream tasks like object detection or segmentation?

---

> ### Author Response · Authors · 2024-11-23
> **Author Response**
>
> ## Weakness 1
> ### **The generalizability of method to practical datasets like clinical and aerial images and evaluating its performance on ViT models.**
> Thanks for the reviewer's valuable comments.
> Our method is specifically designed for large-scale dataset distillation. We conduct experiments on the full-sized, original version of ImageNet-1K.
> We also conduct experiments on the aerial datasets AID[1] to show the effectiveness of our proposed method. The experiments are conducted under the settings of AID with IPC 1, 5, and 10. The performance results and the required extra storage costs are as follows (here, we adopt online soft label generation and regard the teacher model as the extra storage costs). The results show the effectiveness of our proposed method across different high-resolution datasets.
>
> |          | **Downstream Acc.-IPC 1** | **IPC 5**       | **IPC 10**      | **Required Extra Storage Costs**                 |
> |----------|---------------------------|-----------------|-----------------|-------------------------------------------------|
> | **RDED** | 27.2 ± 0.8               | 61.1 ± 0.3     | 78.9 ± 0.2     | 42.69MB                                         |
> | **Ours** | 28.8 ± 0.5               | 65.6 ± 0.8     | 79.2 ± 0.2     | 2.8MB (0.065× of the original extra storage)    |
>
> We have conducted cross-architecture evaluation experiments on Swin-V2-Tiny under the ImageNet-1K setting with IPC 10, as presented in the paper. Additionally, we perform supplementary experiments on ViT-B-16 under the same settings. The evaluation results indicate that our proposed method demonstrates a stronger generalization ability on transformer-based models. We will include the cross-architecture evaluation results on ViT-B-16 in the updated version of the paper.
>
> |          | **ViT-B-16**       | **Swin-V2-Tiny**  |
> |----------|--------------------|-------------------|
> | **RDED** | 19.1 ± 0.4         | 17.8 ± 0.1        |
> | **Ours** | 23.3 ± 0.3 (+4.2)  | 29.5 ± 0.1 (+11.7)|
>
> [1] Gui-Song Xia, Jingwen Hu, Fan Hu, Baoguang Shi, Xiang Bai, Yanfei Zhong, Liangpei Zhang, and Xiaoqiang Lu. Aid: A benchmark data set for performance evaluation of aerial scene classification. IEEE Transactions on Geoscience and Remote Sensing, 55(7):3965–3981, 2017.
>
> ## Weakness 2
> ### **An analysis of distillation costs (memory usage and training time).**
> Thanks for the reviewer's valuable questions and comments. A detailed comparison of the training costs between the proposed method and state-of-the-art methods is indeed of great importance, and we will include this result in the updated version of the paper. Here, we conduct experiments under the ImageNet-1K IPC 10 setting. All methods are evaluated based on the official code, and all experimental configurations, including hyperparameters, are set according to the official default values provided by the authors. Specifically, we measure the time required for each single downstream training iteration as well as the overall memory cost. All experiments are conducted on one single NVIDIA RTX A5000 GPU.
> |              | **Ours**  | **SRe²L**       | **G_VBSM**     | **RDED**        |
> |--------------|-----------|-----------------|----------------|-----------------|
> | **Runtime**  | 0.21s     | 0.75s          | 0.77s          | 0.12s           |
> | **Peak Memory** | 4004MiB  | 20850MiB       | 23310MiB       | 4538MiB         |
>
> ## Weakness 3
> ### **The paper lacks comparisons with key state-of-the-art methods such as DREAM, DataDAM, and SeqMatch in Table 2.**
> Thanks for the reviewer's questions and suggestions. Our method specifically addresses the issue of high storage costs associated with soft labels in large-scale dataset distillation, such as the full-sized ImageNet-1K. In contrast, methods like DREAM, DataDAM, and SeqMatch primarily focus on the distillation of smaller datasets, such as CIFAR-10. Due to computational resource constraints, these approaches face challenges in scaling up to larger datasets, such as the full-sized ImageNet-1K. Moreover, our proposed method has already been compared with the current state-of-the-art methods for large-scale dataset distillation in the paper.

---

> > ### Author Response · Authors · 2024-11-23
> > **Author Response-2**
> >
> > ## Question 1
> > ### **Cross-architecture generalization capability on ViT models.**
> > We have conducted cross-architecture evaluation experiments on Swin-V2-Tiny under the ImageNet-1K setting with IPC 10, as presented in the paper. Additionally, we perform supplementary experiments on ViT-B-16 under the same settings. The evaluation results indicate that our proposed method demonstrates a stronger generalization ability on transformer-based models. We will include the cross-architecture evaluation results on ViT-B-16 in the updated version of the paper.
> > |          | **ViT-B-16**       | **Swin-V2-Tiny**  |
> > |----------|--------------------|-------------------|
> > | **RDED** | 19.1 ± 0.4         | 17.8 ± 0.1        |
> > | **Ours** | 23.3 ± 0.3 (+4.2)  | 29.5 ± 0.1 (+11.7)|
> >
> > ## Question 2
> > ### **Performance on higher IPCs.**
> > Thanks for the reviewer's valuable question. Yes, HeLlO can maintain strong performance at higher IPCs. Here, we conduct experiments under the settings of ImageNet-100 with IPC 100. The performance results and the required extra storage costs are as follows (here, we adopt online soft label generation and regard the teacher model as the extra storage costs):
> > |          | **Downstream Accuracy** | **Required Extra Storage Costs**                  |
> > |----------|--------------------------|---------------------------------------------------|
> > | **RDED** | 75.9 ± 0.1              | 42.83MB                                           |
> > | **Ours** | 76.2 ± 0.2              | 10.20MB (0.24× of the original extra storage)    |
> >
> > ## Question 3
> > ### **How does the proposed framework perform on other downstream tasks like object detection or segmentation?**
> > Thanks for the reviewer's valuable question. Our proposed method can also be seamlessly adapted to segmentation or detection tasks. In these tasks, we leverage foundation models pre-trained on large-scale datasets as base models, employing LoRA-Like Knowledge Transfer methods to adapt to downstream tasks. Our proposed method offers several benefits: 1) The foundation models are pre-trained on massive datasets, enabling easy adaptation and generalization to various target datasets; 2) these models contain rich, applicable knowledge that can be effectively leveraged during the training process; 3) being open-source, they do not require additional storage space and can be accessed on demand; 4) by using LoRA-like methods, we only need to store low-rank matrices instead of the entire large model, reducing storage requirements while maintaining adaptability to different datasets without the need for full retraining.
> > Here, we also conduct experiments on semantic segmentation tasks, here we follow the definition and setting of dataset distillation, utilizing a very small subset of 10,000 images from the SA-1B dataset (around 0.09\% of the original dataset) for downstream model training. Here, the baseline performance is evaluated on the 89.85M SAM-B model (storage costs 358.32MB) to generate soft labels for the downstream tasks. Our proposed method adopts the SAM-B model as the base model and applies our proposed label space lightening strategies, finally with only 5.64M (storage costs 22.56MB, 6.3\% of the original storage costs) learnable and required to store parameters. The baseline performance is mIoU 49.46\%, while our proposed method, uses only 6.3\% storage costs, and achieves better performance with mIoU of 50.75\%.
> > |                          | **Downstream mIoU** | **Required Extra Storage Costs**                  |
> > |--------------------------|---------------------|---------------------------------------------------|
> > | **SAM-B Directly Guided**| 49.46%             | 358.32MB                                          |
> > | **Ours**                 | 50.75%             | 22.56MB (0.063× of the original extra storage)    |
> >
> > Performance on semantic segmentation tasks. Here, SAM-B Directly Guided refers to directly adopting the SAM-B model to generate the soft labels for downstream tasks. Our method utilizes SAM-B as the base model with only 6.3\% of the original costs to achieve better performance.

---

> ### Author Response · Authors · 2024-11-25
> **Author Response-3**
>
> We greatly appreciate the reviewer’s comprehensive review. We hope our previous responses have satisfactorily addressed all concerns. Should any further questions arise or if there are topics that require more detailed discussion, we remain fully available and would be pleased to continue the discussion.

---

> > ### Comment · Reviewer_1rVR · 2024-11-25
> >
> > Thank you to the authors for the clarification and detailed explanation. The insights provided are valuable and well-reasoned. While some weaknesses, such as W3, are not valid—since works like DataDAM have successfully performed dataset distillation on large-scale datasets like ImageNet-1K—however, I will raise my initial score to 6, as most of my concerns have been addressed.

---

> > > ### Author Response · Authors · 2024-11-28
> > > **Author Response-4**
> > >
> > > We sincerely appreciate the reviewer's thoughtful and constructive feedback. Based on the reviewer's valuable comments, we have made the following revisions in our paper:
> > > 1. To demonstrate the generalization of our proposed method, we have conducted experiments on the aerial dataset AID, achieving superior performance compared to the previous SOTA with reduced storage requirements. We have added the experimental details and results in Section 4.6 of the main text, and highlighted them for ease of reference by the reviewer.
> > > 2. We have conducted a comparison of the runtime (per iteration) and peak memory during downstream training for our method compared with the previous SOTA methods. The experimental details and results have been added to Section B.3 of the supplementary material, and highlighted for the reviewer's convenience.
> > > 3. We have conducted cross-architecture experiments on ViT architectures, and have added the experimental details and results in Section B.2 of the supplementary material. The additions have been highlighted for ease of reference.
> > > 4. To demonstrate the performance of our method, we have conducted experiments on ImageNet-100 with IPC 100. The results have been added to Section B.1 of the supplementary material and highlighted for ease of reference.
> > > 5. In addition, to demonstrate the generalization of our proposed method, we have successfully adapted our method to segmentation tasks. The experimental details and conclusions have been added to Section B.4 of the supplementary material and highlighted for ease of reference by the reviewer.
> > >
> > > We would like to extend our sincere and heartfelt gratitude to the reviewer for the considerable time and effort invested in evaluating our work, as well as for the detailed and constructive feedback provided.

---

### Official Review · Reviewer_PbPw · 2024-11-04

**Soundness:** 3
**Presentation:** 2
**Contribution:** 2
**Rating:** 6
**Confidence:** 5

**Summary:**

This paper proposes an interesting alternative to label storage in the context of data distillation. By developing image-to-label projectors based on existing works in CLIP and LoRA, their proposed method, HeLlo can dramatically reduce the original storage requirements of the soft label set by generating them in an online fashion. Understandably with large datasets (i.e. large number of classes), the size of softlabels will scale linearly and can create storage bottlenecks for further downstream usage of the distilled data. Experimentally, we see that the proposed method can far surpass the performance of SoTA work RDED on the distillation benchmarks -- including a variety of different downstream architectures (both convolutional and transformer).

**Strengths:**

I do think this paper approaches a problem that isn't really being avidly researched in the field of data distillation. In particular, most works aim to reduce the costs associated with the number of images, or the learning paradigms for distillation. However, this work looks at an add on to existing distillation techniques in an effort to reduce the storage requirements of the labels. From a novelty perspective, I would agree that this paper incorporates new or innovative techniques into solving a unique problem. I believe there is value in the idea of an image-label-projector, particularly in distillation, and it is interesting to see how CLIP and LoRA have influenced the author's designs, showing a blend of recent developments in adjacent domains. The results are quite convincing, and the ablations studies appear throughout, including downstream applications, such as continual learning. I believe this contribution has strengths primarily in novelty and target scope. However, the latter is also a weakness I elaborate on below.

**Weaknesses:**

Despite the interesting approach taken in this paper, I find there to be a few crucial weaknesses that may overshadow the benefit of the approach. Data Distillation computational costs are often computed regarding the size/number of images as these often take up more storage space than the labels. Understandably at lower image/class ratios, this distribution may deviate (as soft labels would remain scaled at the number of classes, say 1K on ImageNet) -- however I am not convinced that the storage cost of soft labels is indeed that significant. In particular, the asses data distillation methods are on Image Classification, hence each image is associated with one vector of soft labels. Thus at a reduced number of images, 1 or even 50 images per class, the incurred cost of soft labels is actually quite small, (appears to be a couple GB's on ImageNet1K at 10 ipc). Finally, I think alternate methods of logit compression should be compared, such as simple quantization of the logits to a lower precision, as they may not indeed cause significant performance degradation, but will greatly reduce the storage costs. I think the primary weakness here is whether the contribution in this paper actually solves a problem that is being faced in data distillation. As currently I think the scope is not actually a commonly faced problem in the field.

Additionally, I think the professionalism/clarity of the figures can be improved, particularly Figure 1.0, where certain words are covered by symbols, etc.

**Questions:**

Some questions I think that could be helpful here:

1. Can we apply this to more complex tasks (like segmentation of vision language models)? This would widely increase the scope of the paper, and understandably storage requirements on these tasks would be significantly more prominent (i.e. segmentation would have softlabels for each pixel so the computational costs would be far larger).

2. How does the method compare with naive compressions such as quantization etc.

I think the biggest things for the authors to focus on during the rebuttal/discussion phase should be comprehensive comparisons against existing storage compression techniques (particularly simple ones, such as quantization or even interpolation). Additionally, I would find this work's significance far more compelling as it has been applied to a task where the storage costs of soft labels are a clear bottleneck, like semantic segmentation.

---

> ### Author Response · Authors · 2024-11-23
> **Author Response**
>
> ## Weakness
> Thanks for the reviewer's valuable questions and suggestions. For current large-scale dataset distillation methods, at each downstream iteration, the synthetic images are augmented using well-defined strategies, such as CutMix. Each augmented image is then assigned a soft label to ensure the accuracy and informativeness of the knowledge transfer for downstream tasks. Consequently, the total number of soft labels is calculated as IPC × number of classes × number of downstream training epochs. In the case of previous methods, the number of training epochs for downstream training is set to 300; therefore, we calculate the storage requirement assuming 300 epochs. This results in a significant storage cost for maintaining the soft labels. Also, there still requires extra storage space to store the information of the augmentation parameters, such as the image ID, flip status, and coords status, for downstream reproduction.
> Thanks for the reviewer's valuable suggestions, we will modify and improve the figures in the new version.
>
> ## Question 1
> ### **Results on more complex tasks.**
> Yes, here, we conducted additional experiments on semantic segmentation tasks. Following the definition of the dataset distillation, we select 0.09\% of the dataset SA-1B, 10,000 images for downstream training.
> Here, the baseline performance is evaluated on the original SAM-B model (89.58M parameters, 358.32MB actual storage) to generate soft labels for the downstream model training. Our proposed method adopts SAM-B as the base model and applies our proposed label space lightening strategy to reduce the storage costs, finally with only 5.64M parameters (actual storage 22.56MB) trained and stored.
> For the baseline performance, the downstream model achieves mIoU of 49.46\%, while our proposed method, with only 6.3\% storage costs, achieves higher performance with mIoU of 50.75\%.
> |                          | **Downstream mIoU** | **Required Extra Storage Costs**                      |
> |--------------------------|---------------------|------------------------------------------------------|
> | **SAM-B Directly Guided**| 49.46%             | 358.32MB                                             |
> | **Ours**                 | 50.75%             | 22.56MB (0.063× of the original extra storage)       |
>
> Performance on semantic segmentation tasks. Here, SAM-B Directly Guided refers to directly adopting the SAM-B model to generate the soft labels for downstream tasks. Our method utilizes SAM-B as the base model with only 6.3% of the original costs to achieve better performance.
>
> ## Question 2
> ### **Comparison with naive compressions such as quantization.**
> We sincerely appreciate the reviewer’s insightful comments. Indeed, a direct comparison involving label quantization is a crucial aspect that warrants careful examination. In response, we have conducted experiments under the ImageNet-1K IPC 10 setting on the NVIDIA RTX A5000 GPU, where we applied precision reduction to the labels for the previous state-of-the-art method and compared the results of our proposed method.
> From the results, it can be observed that there is no significant decline in downstream performance when reducing the precision to FP16 (50\%) or INT8 (25\%). However, a noticeable decline occurs when the precision is further reduced to INT4 (12.5\%), which represents the limit for logit quantization. Despite this, we are able to compress storage to less than 0.1\% with comparable performance.
>
> |                          | **FP32**         | **FP16**          | **INT8**         | **INT4**         | **Ours**          |
> |--------------------------|------------------|-------------------|------------------|------------------|-------------------|
> | **Downstream Acc.**      | 42.0 ± 0.1      | 42.2 ± 0.2       | 41.2 ± 0.1      | 39.3 ± 0.1      | 43.7 ± 0.1       |
> | **Memory Costs**         | 11596.6MB       | 5798.3MB (0.5×)   | 2899.2MB (0.25×) | 1449.6MB (0.125×)| 3.3MB (0.0003×)    |

---

> ### Author Response · Authors · 2024-11-25
> **Author Response-2**
>
> We are truly grateful for the reviewer’s valuable time and constructive feedback. We hope our response has satisfactorily resolved the concerns raised. Should there be any additional questions or areas where more clarification is required, we would be pleased to engage in further discussion and provide any additional information.

---

> ### Author Response · Authors · 2024-11-28
> **Author Response-3**
>
> We are truly grateful for the reviewer's constructive comments. We have implemented the following revisions in the paper according to the reviewer's feedback:
> 1.  We have conducted a comparison with the logits quantization strategies, and have added the experimental details and results in Section 4.5 of the main text. The additions have been highlighted in red for the reviewer's convenience.
> 2. We have successfully adapted our method to segmentation tasks, and have included the experimental details and results in Section B.4 of the supplementary material. The additions have been highlighted in red for the reviewer's convenience.
> 3. We have revised Table 1 and provided additional explanations to make the importance of the topic being addressed in our paper more clear.
> 4. We thank the reviewer for their attention to detail. We have revised Figure 1 to ensure that the text is not covered by the symbols, making it more professional.
>
> We sincerely thank the reviewer once again for the reviewer's comments and suggestions. We hope that our responses have addressed the reviewer's concerns, and we look forward to further discussions with the reviewer.

---

### Official Review · Reviewer_LdWz · 2024-11-04

**Soundness:** 1
**Presentation:** 1
**Contribution:** 2
**Rating:** 5
**Confidence:** 3

**Summary:**

This work introduces HeLlO (Heavy Labels Out), a framework designed to address the challenge of high storage requirements for soft labels in large-scale dataset distillation. Instead of storing the entire soft label, an off-the-shelf pre-trained feature extractor is employed to generate labels in an online fashion during downstream network training. This approach significantly reduces the storage burden, especially when dealing with extensive datasets that necessitate numerous soft labels. Moreover, to mitigate the distribution gap between the target label space and the label space inherent to the pre-trained model, a LoRA-like fine-tuning strategy is applied.

**Strengths:**

The work effectively tackles the critical issue of high storage requirements for soft labels.

**Weaknesses:**

- The comparison presented in Table 1 appears unfair. Unlike the baseline methods, the proposed method requires storing a teacher model (or its low rank version), which complicates a direct comparison. It is unclear why the storage size of the teacher model is relevant here. Please provide a more comprehensive comparison that includes the storage requirements for both the teacher model and the labels across all methods.
- Table 1 contains numerous missing values, which hinders a comprehensive comparison. The reviewer encourages the authors to provide complete results. At a minimum, please include the IPC 50 results for SRe2L and G-VBSM on the Places365-Standard dataset, as the proposed method performs lower than the baseline in this setting.
- Poor writing, there are many typos and ambiguous notations in the proof of Proposition 1, e.g. both $r_i$ and $r^{(i)}$ represent the text description for class $i$. Detailed feedback on these issues is provided in the questions part.

**Questions:**

- Ambiguous notations
    1. In line 198, $r_i$ is described as the text description for class $i$. What is $r^{(i)}$ in Equation (2)?
    2. In line 199, $v_T$ is used for the text description of the $i$-th class. What is the meaning of $v_T^{(i)}$ in Equation (3)?
    3. In line 201, $W$ is defined as a matrix of size $d_f \times C$, yet the authors also write $W = \textbraceleft w^{(i)} \textbraceright_{i=0}^{C-1}$, which suggests $W$ is a set. Could the authors clarify this discrepancy?
- What is the definition of weak teacher in line 237? Does is differ from the teacher defined in line 162?
- The reviewer is curious about the potential outcomes of using a model pre-trained on the original real dataset as the label projector. Could the authors provide more insights or results related to this scenario?

---

> ### Author Response · Authors · 2024-11-23
> **Author Response**
>
> ## Weakness 1
> ### **Provide a more comprehensive comparison for the storage requirements across all methods.**
> Thanks for the reviewer's valuable suggestions. Our intention is to illustrate the storage under different soft label generation strategies. We apologize for any confusion this may have caused. The current state-of-the-art large-scale dataset distillation methods for soft label generation can be categorized into two main approaches: one involves pre-generating and storing the labels (e.g., Yin et al. (2024) and Shao et al. (2024)), while the other generates labels online during downstream training (e.g., Sun et al. (2024)). Our method primarily falls into the second category. Specifically, for the first strategy, the storage costs are from two parts. 1) The labels for each augmented image. Here, for each iteration, the synthetic images will apply well-defined augmentation strategies, e.g., CutMix, and each augmented image should be assigned with soft labels. 2) The information of the augmentation, e.g., the image ID, flip status, coords status for CutMix, for downstream reproduction. The other approach generates soft labels dynamically during the training process, so the additional storage required is limited to the teacher model. We will improve the presentation of Table 1 in the updated version to provide a clearer comparison.
> The actual storage costs comparison is as follows:
> | **Datasets**            |  IPC  | **SRe²L**    | **G_VBSM**  | **RDED**     | **Ours**    |
> |--------------------------|---|--------------|-------------|--------------|-------------|
> | **ImageNet-100**         | 1 | 6.9MB       | -           | 42.8MB       | 2.6MB       |
> |                          | 10| 64.8MB      | -           | 42.8MB       | 2.6MB       |
> |                          | 50| 324.2MB     | -           | 42.8MB       | 2.6MB       |
> | **Places365-Standard**   | 1 | 79.3MB      | -           | 43.4MB       | 3.0MB       |
> |                          | 10| 790.4MB     | -           | 43.4MB       | 3.0MB       |
> |                          | 50| 3950.6MB    | -           | 43.4MB       | 3.0MB       |
> | **ImageNet-1K**          | 1 | 579.8MB     | 582.2MB     | 44.7MB       | 3.3MB       |
> |                          | 10| 5798.3MB    | 5821.5MB    | 44.7MB       | 3.3MB       |
> |                          | 50| 28990.8MB   | 29110.6MB   | 44.7MB       | 3.3MB       |
>
> ## Weakness 2
> ### **Reproduce for the baseline methods on Places365-Standard.**
> Thanks for the reviewer's valuable suggestions. We reproduce the results for SRe2L on the Places365-Standard, the results are shown as follows. For G-VBSM, since it requires preparing a model pool of well-trained models with different architectures on the target dataset and selecting the most suitable model architecture combination, we are still in the process of preparation and evaluation. We will update the paper immediately once the results are available. The evaluation results show that our proposed method surpasses the SRe$^2$L for all different IPC settings on the Places365-Standard dataset.
> |                     | **IPC 1**         | **IPC 10**       | **IPC 50**       |
> |---------------------|-------------------|------------------|------------------|
> | **SRe²L**          | 1.4 ± 0.2        | 9.5 ± 0.1        | 31.3 ± 0.1       |
> | **Ours**           | 7.1 ± 0.1        | 33.4 ± 0.3       | 41.2 ± 0.1       |
>
> ## Weakness 3 & Question 1
> ### **Ambiguous notations.**
> Thanks very much for the reviewer's thorough review. We apologize for any misunderstanding caused by the way we presented our formulas. We will make corrections in the next version.
> 1) We are sincerely sorry for this typo, $r_i$ here should represent the same as $r^{(i)}$, the text description of the class $i$.
> 2) We again apologize for the typo, and any misunderstanding caused. Line 199 should be "$v_I\in\mathbb{R}^{B\times d_f}$ and $v_T\in \mathbb{R}^{C\times d_f}$ refer to the normalized embedding for the input image and the text description, $v_T^{(i)}$ is for $i^{th}$ class."
> 3) Here, we intend to represent it as $W=[w^{(0)}, w^{(1)}, \dots, w^{(C-1)}]$, we are so sorry for the unclear representation, and we will correct all of the above ambiguous notations in the new version. Thanks again for the reviewer's thorough and careful reviews.

---

> > ### Author Response · Authors · 2024-11-23
> > **Author Response-2**
> >
> > ## Question 2
> > ### **The definition of weak teacher in line 237.**
> > Thanks for the reviewer's valuable questions. There is a difference here. The teacher model defined in line 162 is directly used for label generation in downstream training. In our case, the teacher model is employed to better align our projector with downstream tasks. We will emphasize and clarify this point in the new version. Additionally, we highlight 'weak' here because the teacher models we adopted are at the early training stage. Please refer to our ablation study in section 4.4.3. The experiment shows that the teacher models at early stage will benefit our projector to have better generalization for downstream tasks.
> > ## Question 3
> > ### **The results on the model pre-trained on the original real dataset as the label projector. More insights or results related to this scenario.**
> > Thanks for the reviewer's valuable questions. We conduct experiments on the ResNet-50, ViT_b_16, CLIP, and DINOv2. The results are shown in the following. We would also like to share two insights here: 1) stronger pre-trained teacher models would not always benefit the downstream tasks; 2) similar network architectures and model sizes of the teachers and the students (for downstream tasks) help to transfer the knowledge to students and improve the performance of the students.
> >
> > |                          | **CLIP-LoRA (Ours)** | **DINOv2-LoRA (Ours)** | **ResNet-50** | **ViT_b_16** | **CLIP** | **DINOv2** |
> > |--------------------------|----------------------|------------------------|---------------|--------------|----------|-----------|
> > | **Downstream Acc.**      | 43.7 ± 0.1          | 44.2 ± 0.1            | 34.9 ± 0.1    | 17.8 ± 0.2   | 28.2 ± 0.2 | 23.3 ± 0.2 |

---

> ### Author Response · Authors · 2024-11-25
> **Author Response-3**
>
> We deeply appreciate the reviewer’s thoughtful insights and hope that our response has resolved the issues raised. Should there be any further questions or points that require more in-depth discussion, we would be honored to continue the conversation and provide any needed clarifications.

---

> > ### Comment · Reviewer_LdWz · 2024-11-25
> >
> > I accept the arguments for W2 and Q3. For W3 and Q1, I believe these are minor typos, and they do not obstruct my understanding. As for W1, I don’t think such a major revision is permissible. For Q2, a clear and proper definition of the proposed components is important to me. Overall, I won’t change my score.

---

> > > ### Author Response · Authors · 2024-11-25
> > > **Author Response-4**
> > >
> > > We sincerely thanks for the reviewer’s thorough review of our manuscript and for providing such valuable feedback. We have now uploaded the revised version of the paper, incorporating all suggested changes. The revisions are highlighted in red for the reviewer’s convenience as follows.
> > > ### **1. Clarity of the Table 1**
> > > We have revised the tables to enhance their clarity and readability, based on the reviewer’s insightful suggestions. The updated versions present the information more effectively, ensuring that all readers can better comprehend the content.
> > > ### **2. Definition Issues for Weak Teachers**
> > > We have clarified and elaborated on the definitions that were previously unclear. Here, the weak teacher refer to the model trained on the original dataset terminated at the early training stage. These additions help to eliminate ambiguity and ensure that the key concepts are more accessible and precise. The revised lines 236-237 are highlighted in red for the reviewer’s convenience.
> > > ### **3. Ambiguous Notations in Proposition 1**
> > > All ambiguous notations in proposition 1 identified have been corrected, with changes in lines 198-202 highlighted in red for transparency and ease of review.
> > >
> > > We greatly appreciate the reviewer’s detailed and constructive feedback, which has significantly contributed to improving the quality of our manuscript. We believe these revisions result in a clearer and more accurate presentation of our research. We remain open to any further suggestions or comments the reviewer may have and are grateful for the reviewer’s continued consideration.
> > > Thanks once again for the reviewer’s time and effort in reviewing our manuscript.

---

> > > > ### Author Response · Authors · 2024-11-28
> > > > **Author Response-5**
> > > >
> > > > We would like to once again sincerely thank the reviewer for the constructive and insightful feedback. Based on the reviewer's suggestions, we have made the following modifications to the paper:
> > > > 1. We have revised Table 1 to enhance its clarity and readability, in response to the reviewer's insightful suggestions. The updated version presents the information more effectively.
> > > > 2. We have clarified and elaborated on the definitions that were previously unclear. Specifically, the term "weak teacher" refers to a model trained on the original dataset terminated at the early training stage. These clarifications help to eliminate ambiguity and make key concepts more accessible and precise. The revised lines 236-237 have been highlighted in red for the reviewer’s convenience.
> > > > 3. We have corrected all ambiguous notations identified in Proposition 1, with modifications highlighted in red in lines 198-202 for transparency and ease of review.
> > > > 4. We explored the applicability of different foundation models to our method, as well as the downstream performance of directly applying various pre-trained models. The relevant content has been added to Section 4.4.4 of the main text, with changes highlighted in red in red for the reviewer’s convenience.
> > > >
> > > > We would like to once again thank the reviewer for the insightful and valuable suggestions. We hope that these revisions have adequately addressed your concerns.

---

### Official Review · Reviewer_KtxV · 2024-11-06

**Soundness:** 3
**Presentation:** 3
**Contribution:** 3
**Rating:** 5
**Confidence:** 3

**Summary:**

This paper addresses the high storage cost of soft labels in dataset distillation that generates synthetic labels directly from images. It leverages open-source models like CLIP and a LoRA-like fine-tuning strategy to extract the labels, bridging the gap between pre-trained and target distributions.

**Strengths:**

1. The motivation of this paper is clear and easy to understand.
2. The proposed method is simple and effective.

**Weaknesses:**

I am convinced of the motivation of this paper, but there are significant concerns on using CLIP to generate labels from images:

1. Why not use stronger supervised models or self-supervised models? If we can use another stronger model like CLIP to generate labels from images, why not use it directly for the target task, or replace CLIP (weakly-supervised) with other models like supervised ViT or self-supervised DINOv2? Authors finetune CLIP vision encoder with LoRA, and text embeddings to adapt the specific dataset, which can be seen as a kind of supervised trained model. If we change it to supervised ResNet, ViT, or self-supervised DINOv2 (versus the performance reported with R18 from scratch), the performance may be better. Similarly, why not finetune the foundation models (CLIP or DINOv2) with LoRA on the distilled datasets if we have already used a foundation model?

2. The effectiveness of the proposed method. The storage of classifier weights is not important compared to labels, only 0.7MB in Table 3. In addition, the improvements of Text-Embedding-Based Init and Image Update are quite marginal (0.1 in accuracy). Most improvements come from LoRA-Like Knowledge Transfer, that is to say, adapting the supervised information of the dataset to CLIP vision encoder. 'M' is a bit confusing, please change 'M' to 'MB' in the paper.

**Questions:**

Why there is no 'Size of Labels' for L Yin et al. (2024) and Sun et al. (2024)

---

> ### Author Response · Authors · 2024-11-23
> **Author Response**
>
> ## Weakness 1
> ### **W1.1 Replace CLIP with stronger supervised models or self-supervised models.**
> Thanks very much for the reviewer's valuable suggestions.
> Here, as mentioned in our paper, we adopt foundation models like CLIP for the following reasons: 1) it has been pre-trained on the massive data, and can readily adapt and generalize to various target datasets; 2) it has rich and feasible knowledge that can be leveraged during the training process; 3) it is open-source, such that it does not require extra storage space and can be accessed on demand.
> Benefiting from the aforementioned characteristics, the transformation to different downstream datasets can be achieved by LoRA, instead of finetuning or re-training the entire model. Also, the storage is significantly reduced as we only need to store some low-rank matrices.
>
> Here, we would like to mention that other foundation models, like DINOv2, are also applicable in our proposed method as an alternative to CLIP for LoRA-Like Knowledge Transfer and other strategies. As shown in the following table, we also conduct the experiments under the settings of ImageNet-1K with IPC 10 for DINOv2 with LoRA-Like Knowledge Transfer.
>
> In other words, we use CLIP in the main manuscript as a proof-of-concept validation of our proposed method. We will also add the experiments on other foundation models for further demonstration.
>
> For other supervised models like ResNet and ViT, we do not adopt them for the following reasons: 1) it can not easily generalize to other datasets, and it may modify the architecture and re-train the whole model to satisfy the downstream requirements; 2) it cannot be accessed easily. As for arbitrary datasets, it can be difficult to find the pre-trained models for the specific target dataset;
> and 3) compared with the very little storage costs as the LoRA required, the storage costs for ResNet and ViT are increasing if we would like to process more datasets.
>
> In addition, directly employing strong models for label generation is not necessarily beneficial for downstream tasks. Here, we utilize stronger models ResNet-50, ViT_b_16, CLIP (directly applied), and DINOv2 (directly applied) for label generation, the results are shown as follows. The experiments are under the settings of ImageNet-1K with IPC 10.
> |                          | **CLIP-LoRA (Ours)** | **DINOv2-LoRA (Ours)** | **ResNet-50** | **ViT_b_16** | **CLIP** | **DINOv2** |
> |--------------------------|----------------------|------------------------|---------------|--------------|----------|-----------|
> | **Downstream Acc.**      | 43.7 ± 0.1          | 44.2 ± 0.1            | 34.9 ± 0.1    | 17.8 ± 0.2   | 28.2 ± 0.2 | 23.3 ± 0.2 |
>
>
> ### **W1.2 Finetune the foundation models (CLIP or DINOv2) with LoRA on the distilled datasets.**
>
> Directly finetuning the foundation models with LoRA on the distilled datasets leads to overfitting, as the distilled dataset is significantly smaller compared to the original datasets. This also presents challenges in generalizing to downstream tasks. We also conduct experiments on CLIP for finetuning with LoRA on the distilled datasets. The evaluation result is 39.8 $\pm$ 0.1 compared with 43.7 $\pm$ 0.1 (finetuning on the original dataset) under the settings of ImageNet-1K with IPC 10.
>
> ## Weakness 2
> ### **The effectiveness of the proposed method.**
> Thanks for the reviewer's valuable suggestions. In our paper, M specifically refers to the number of the parameters. We apologize for any potential misunderstanding this may have caused. We will revise the text to clarify this point and include the evaluation results for storage.
> Here, the actual storage for the classifier (around 4MB) is quite significant compared to the storage of the synthetic images (~50\% of the storage of the images of ImageNet-1K with IPC 1). Moreover, the goal of our method is to achieve performance comparable to the state-of-the-art method, while using extremely little storage costs. Thus, a 4MB reduction in storage is quite important in our method. More importantly, in this context, Text-Embedding-Based Init accelerates the convergence and enhances the training speed of the projector. We will include metrics such as the loss during the training process in our updated version for reference.
> For the Image Update, its effectiveness varies under different settings. For instance, on the ImageNet-1K dataset with IPC 1, it demonstrates more significant improvements, achieving a gain of 1.0 (12.9 $\pm$ 0.3 with image update strategy compared with 11.9 $\pm$ 0.1 without image update strategy). Overall, it contributes positively to the downstream tasks. However, since our projector training phase narrows the distance between the observer model and the projector, the downstream performance gains may exhibit some variation.

---

> ### Author Response · Authors · 2024-11-23
> **Author Response-2**
>
> ### **(Cont.) The effectiveness of the proposed method**
> **The ablation studies under the settings of ImageNet-1K with IPC 1**
> |                                   | **Probe Linear CLIP** | **+ Multi-Weak-Teacher Guided** | **+ LoRA-Like Knowledge Transfer** | **+ Text-Embedding-Based Init.** | **+ Image Update** |
> |-----------------------------------|-----------------------|----------------------------------|-------------------------------------|-----------------------------------|---------------------|
> | **Acc.-IPC 1** (ImageNet-1K)      | 5.3 ± 0.1            | 6.4 ± 0.2 (+1.1)               | 11.4 ± 0.2 (+5.0)                  | 11.9 ± 0.1 (+0.5)                | 12.9 ± 0.3 (+1.0)   |
>
>
> ## Question 1
> ### **Why there is no 'Size of Labels' for Yin et al. (2024) and Sun et al. (2024)**
>
> Thanks very much for the reviewer's insightful question. Yin et al. (2024) and Shao et al. (2024) employ a soft label generation strategy involving pre-generating and storing labels prior to downstream training, and Sun et al. (2024) generates soft labels online during downstream training. The pre-generation approach involves additional storage requirements in two aespects: (1) Soft labels for each augmented image: During each epoch, synthetic images will perform predefined augmentation strategies, such as CutMix, and each augmented image is assigned a corresponding soft label. (2) Augmentation details for downstream reproduction, e.g., image ID, flip status, and coordinates for CutMix to ensure reproducibility.
> The online generation strategy requires storing the teacher model(s). Table 1 presents a comparison of the storage requirements for soft labels in Yin et al. (2024) and Shao et al. (2024), along with the teacher model storage requirement for Sun et al. (2024). We sincerely apologize for any confusion that may have resulted from our initial presentation. In the revised version of our paper, we update Table 1 to clearly indicate the actual total storage requirements for each method, aiming to ensure better presentation to prevent any further misunderstandings.
> | **Datasets**            |   | **SRe²L**    | **G_VBSM**  | **RDED**     | **Ours**    |
> |--------------------------|---|--------------|-------------|--------------|-------------|
> | **ImageNet-100**         | 1 | 6.9MB       | -           | 42.8MB       | 2.6MB       |
> |                          | 10| 64.8MB      | -           | 42.8MB       | 2.6MB       |
> |                          | 50| 324.2MB     | -           | 42.8MB       | 2.6MB       |
> | **Places365-Standard**   | 1 | 79.3MB      | -           | 43.4MB       | 3.0MB       |
> |                          | 10| 790.4MB     | -           | 43.4MB       | 3.0MB       |
> |                          | 50| 3950.6MB    | -           | 43.4MB       | 3.0MB       |
> | **ImageNet-1K**          | 1 | 579.8MB     | 582.2MB     | 44.7MB       | 3.3MB       |
> |                          | 10| 5798.3MB    | 5821.5MB    | 44.7MB       | 3.3MB       |
> |                          | 50| 28990.8MB   | 29110.6MB   | 44.7MB       | 3.3MB       |

---

> ### Author Response · Authors · 2024-11-25
> **Author Response-3**
>
> We sincerely appreciate the reviewer’s time and thoughtful feedback. We hope our response has addressed your concerns. If there are any additional questions or areas that could benefit from further clarification, we would be more than happy to provide a detailed response and engage in further discussion.

---

> > ### Author Response · Authors · 2024-11-28
> > **Author Response-4**
> >
> > We would like to express our sincere gratitude to the reviewer for the constructive comments. Based on the reviewer's valuable comments, we have made the following modifications in the revised manuscript:
> > 1. We have explored the applicability of our method to different foundation models, e.g., DINOv2, as well as the impact of different pre-trained models (particularly strong models, e.g., ResNet-50 and ViT-B-16) on the downstream training performance. We have added this content to Section 4.4.4 of the paper, and highlighted it in red for the reviewer’s convenience.
> > 2. We have added the progression of training loss, training accuracy, and test accuracy with and without the Text Embedding Init. step during downstream training, in order to demonstrate the effect of this step in accelerating convergence. These results have been added to Figure 2 and highlighted in red for ease of reference.
> > 3. We have included the results of an ablation study under the ImageNet-1K IPC 1 setting in Table 3, highlighted in red for ease of reference, to more comprehensively demonstrate the effectiveness of each step of our method.
> > 4. In addition, we have also revised Table 1 to present the storage requirements for each method more clearly and avoid the confusion about 'M' and 'MB'.
> >
> > We sincerely thank the reviewer for the insightful comments. We hope these responses have addressed the reviewer's concerns, and we are more than happy to have the discussion with the reviewer.

---

> ### Author Response · Authors · 2024-12-01
> **A Sincere Request to Check Our Responses**
>
> Dear Reviewer KtxV,
>
> We would like to inquire whether our previous responses have adequately addressed your concerns. We would greatly appreciate any further feedback from you and look forward to your response. If you have any additional questions or require more information, please do not hesitate to let us know. We would be more than happy to discuss further and provide any additional details needed.
>
> Thanks very much for the your valuable time and suggestions regarding our work.

---

> > ### Author Response · Authors · 2024-12-02
> > **Sincerely Looking Forward to Your Review and Reply to Our Responses**
> >
> > Dear Reviewer KtxV,
> >
> > We sincerely appreciate your comprehensive and thoughtful review of our paper. Your suggestions is valuable and insightful, and we have carefully considered your comments and provided detailed responses.
> >
> > With less than one day remaining in the discussion phase, we sincerely hope to know if our responses have adequately addressed your concerns. If you have any additional questions or require further clarification, explanation, and discussion, please do not hesitate to let us know. We would be more than happy to provide any necessary details and engage in further discussions.
> >
> > Thank you once again for your valuable time and efforts in reviewing our work.

---

> > > ### Author Response · Authors · 2024-12-03
> > > **Sincere Request for Your Reply to Our Responses**
> > >
> > > Dear Reviewer KtxV,
> > >
> > > Thank you very much for taking the time to review our work and provide your valuable comments. As the discussion period is approaching its end, we would like to know if our previous response has addressed your concerns. We truly value your comments and would greatly appreciate the opportunity to discuss any possible remaining questions or clarifications. We look forward to your response and hope to engage in further discussion.
> > >
> > > Thank you again for your time and consideration.

---

> > > > ### Author Response · Authors · 2024-12-04
> > > > **Summary of Our Responses**
> > > >
> > > > We sincerely appreciate the reviewer's valuable comments. We have carefully addressed all concerns and would like to summarize our responses as follows:
> > > >
> > > > - **Projector Choice**: We clarify our choice of the foundation model as the base projector and apply our method to other foundation models like DINOv2, demonstrating its effectiveness. We also compare against stronger models as projectors and LoRA-like knowledge transfer to distilled dataset as suggested, showing superior performance and efficiency of our method.
> > > > - **Method Effectiveness**: We add training loss, training accuracy, and test accuracy progression, with and without Text Embedding Init. step, to highlight its role in accelerating convergence. We also include an ablation study under the ImageNet-1K IPC 1 setting to demonstrate each step's effectiveness.
> > > > - **Clearer Representation**: We revise multiple places in the manuscript and ensure that all the definitions, claims, and explanations are clear.
> > > >
> > > > We kindly hope that the reviewer would consider reviewing our responses, and we sincerely hope that they address the reviewer’s concerns.

---

### Meta-Review · Area_Chair_4sEN · 2024-12-20

**Metareview:**

The paper introduces an innovative method to tackle the high storage costs related to soft labels in dataset distillation by employing open-source models and a fine-tuning strategy to derive labels directly from images. However, the primary concerns revolve around the reliance on CLIP for label generation and doubts about the comparative effectiveness of the proposed method when pitted against more robust supervised or self-supervised models. In addition, the paper's contribution is considered marginal, with minimal gains in accuracy and an emphasis on classifier weights storage that may not be as significant as portrayed.

**Additional Comments On Reviewer Discussion:**

I appreciate the efforts made by the authors in addressing the reviewers' concerns during the rebuttal phase. While some reviewers have gained a better understanding of the methodology and have slightly increased their scores, significant concerns have been raised by Reviewer KtxV regarding the effectiveness of the proposed method. Additionally, Reviewer LdWz is worried about the presentation issues. These issues weaken the quality of this work for ICLR.

---

### Decision · Program_Chairs · 2025-01-22

Reject